# Inequality in electricity consumption and economic growth: Evidence from a small area estimation study

Cuong Viet Nguyen[1,2,3], Khuong Duc Nguyen [3,4], Tuyen Quang Tran [1]*

1 Faculty of Economics and Management, International School, Vietnam National University, Hanoi, Vietnam, 2 Mekong Development Research Institute, Hanoi, Vietnam, 3 IPAG Business School, Paris, France, 4 Faculty of Finance and Accounting, Prague University of Economics and Business, Prague, Czech Republic

* tuyenisvnu@gmail.com, tuyentq@vnu.edu.vn

## Abstract

Our study uses a small area estimation method to estimate the average and inequality of per capita kWh consumption for small areas in Vietnam. It shows evidence of a large spatial heterogeneity in the electric power consumption between districts and provinces in Vietnam. Households in the mountains and highlands consumed remarkably less electricity than those in the delta and coastal areas. Notably, we find a U-shaped relationship between the inequality of electricity consumption and economic levels in Vietnam. In poor districts and provinces, there is very high inequality in electricity consumption. Inequality is lower in middle-income districts and provinces.

## 1. Introduction

One of the notable facts about the long-term process of economic development is the Kuznets curve, also known as the inverted U-shape patterns of inequality [1]. Using both time series data of the United States, the United Kingdom and two states in Germany, Kuznets found that as countries developed, inequality first increased, peaked and then declined [2]. Although there have been an intensive number of studies examining the empirical validity of the Kuznets curve for different countries and datasets, the empirical evidence is rather inconclusive [3–7]. For example, while historical investigations from Western European countries such as England, France, Germany, and Sweden tend to support the Kuznets hypothesis, inequality has consistently experienced falls during the process of economic development in some Asian countries such Japan, South Korea and Taiwan [1]. Gallup noted, from a careful review of previous studies, that the Kuznets hypothesis seems to be validated only with cross sectional data, while it should hold within countries [8]. Using an international panel inequality data of 87 countries, the author estimated the relationship between inequality and income from two non-parametric models (i.e., a non- parametric fixed effects trend and a stochastic kernel model) and finds anti-evidence of the Kuznets because inequality decreases in low-income countries and increases in high-income countries. A more recent study by Jovanovic questions the

**Funding:** The authors received no specific funding for this work.

**Competing interests:** The authors have declared that no competing interests exist.

conditions under which the Kuznets curve would exist for a sample of 26 ex-socialist countries in Eastern Europe over the period 1990–2011 and shows evidence that this hypothesis is only valid when there are effective control of companies' power and high taxes [9]. Anwar et alconduct a bibliometric review of research on the EKC based on 2218 articles and 55,051 references from 328 journals further points out that the focus was mostly on developing countries and no consensus really exists regarding the validity of the EKC in terms of existence, shape, and turning points [4].

Inequalities are often measured and discussed in terms of income or some related monetary aspects [10, 11]. Nevertheless, disparities in energy access and use by households within a country migh well reflect inequality in living standards [12]. The residential demand for electricity is also growing consistently in line with the societies' rising economic affluence [12–17]. Thus, the household sector accounts for a large and increasing share of total electricity use and the same trend has been observed in Vietnam [18]. A distinctive economic attribute of electricity demand by households is that it is not determined by the utility for the use of electricity per se. Rather, electricity is a vital input for using a wide range of services provided by many appliances and devices. Therefore, the demand of electricity is driven by the need for services required for a lot of necessities such as cooking, lighting, cooling and heating, to normal and luxury leisure activities [15]. To the extent that a household's income rises, its demand for and ultimately its spending on electricity tend to increase. Income elasticity of electricity demand can change with income levels. This leads, in turn, to inequality in electricity consumption among households, and this inequality can change across different development levels.

Together with economic growth, consumption of electrical energy has increased remarkably in Vietnam. Vietnam's power consumption has increased by more than 20 times over the last three decades [19]. Access to electricity remained however unequal by income group and by region. According to the General Statistics Office of Vietnam (GSO), around 98% of the richest households—the top quintile—had access versus 84.8% for the poorest—the bottom quintile [20]. Disparity was also found among geographic regions. In rich regions such Red River Delta and Southeast nearly 100% of house have access to electricity, while in the poor region such as North West and the Central Highlands were the worst off, with power available to only 72.1% and 87.4% of households, respectively [18]. Compared with inequality in consumption expenditure, inequality in electricity consumption is much higher. In 2010, the Gini index of per capita expenditure was 0.39, while that of per capita electricity consumption was 0.52.

How to reduce poverty and inequalities is among the key missions of the United Nations sustainable development goals [12, 21]. (Such goals also maintain a crucial role in Vietnam's poverty alleviation strategy. In this study, electricity consumption is employed as a proxy to measure living standards. To the best of our knowledge, there is limited evidence on inequalities in electricity in the literature, and no evidence has been provided in Vietnam thus far. A better understanding of the link between energy and inequality would help better design the electricity access policy to ultimately foster sustainable development goals [12].

In this study, we address the inequality-income nexus by investigating, firstly, the spatial distribution of electricity consumption and subsequently the relationship between inequality in electricity consumption and well-being as measured by expenditure consumption per capita across districts and provinces in Vietnam. At the empirical stage, we draw evidence from district-level data which are computed from the 2010 Vietnam Household Living Standard Survey and the 2009 Vietnam Population and Housing Census. Our results indicate that the economic development proxied by per capital expenditure is significantly correlated with power consumption inequality in a nonlinear manner according to a U-shaped curve. The inequality was also found to exist across both provinces and districts. It is particularly

accentuated in poor regions such as the Northern mountains, the Central Highlands, and the Mekong Delta region.

A challenge in estimating the spatial pattern of electricity consumption (and per capita aggregate expenditure) is the lack of data on electricity consumption at the district level. In Vietnam, sample household surveys such as the 2010 Vietnam Living Standard Surveys (henceforth referred to as the 2010 VHLSS) contains data on electricity consumption (and per capita expenditure) of households, but these surveys are not representative at the district or provincial level. On the other hand, population censuses such as the 2009 Population and Housing Census of Vietnam (henceforth referred to as the 2009 VPHC) have the large coverage of households but do not contain data on electricity consumption (and per capita expenditure). In this study, we combine the 2010 VHLSS and the 2009 Population and Housing Census of VietnamVPHC and use the small area estimation method developed by Elbers et alin order to predict the average and inequality of electric power consumption at the district level [22, 23]. We first use the 2010 VHLSS to construct a model of electricity consumption as a function of household and community variables available in both the 2010 VHLSS and the 2009 VHPC. Then, the parameter estimates from this model are applied to the 2009 VHPC to predict the electricity consumption of all households in the population. These household-level data allow us to estimate the mean and inequality of electricity consumption for provincial and districts. Similarly, we follow the same approach to estimate the average and inequality of per capita expenditure for provincial and districts.

The rest of this paper is structured as follows. Section 2 describes the data used and briefly presents a descriptive analysis of electricity consumption in Vietnam. Section 3 introduces the estimation methods. Section 4 reports and discusses the empirical results. Section 5 provides some concluding remarks.

## 2. Data and descriptive analysis

### 2.1. Data

We use two datasets in this study. The first data set is the 15-percent sample of the Vietnam Population and Housing Census (VPHC) conducted by the General Statistics Office of Vietnam in April 2009. To estimate electricity consumption at small areas, we need to use census data. Vietnam Population and Housing Censuses are conducted every 10 years and we were able to have access to the 2009 census at the time of conducting this research. This census contains detailed data on individuals and households. Individual data include information on demographics, education, employment, disability, and migration. Household data includes information on durable assets and housing conditions. The 15-percent sample is representative at the district level. The census covered 3,692,042 households with 14,177,590 individuals. The 2009 VPHC has a large coverage of households, but it does not contain data on electricity consumption of households.

The second dataset is the 2010 Vietnam Household Living Standard Survey (VHLSS). The VHLSSs have been conducted by the GSO with technical support from the World Bank every two years since 2002. VHLSSs contains detailed data on individuals, households and communes. Individual data consist of information on demographics, education, employment, health, and migration. Household data are on durables, assets, production, income and expenditure, and participation in government programs. Unlike the previous VHLSSs, the 2010 VHLSS contains data about not only expenditure on electricity but also the number of kilowatt hours (kWh) that households consumed in the last month. The 2010 VHLSS covered 9,399 households.

In this study, we use the 2010 VHLSS, since it is closer to the 2009 VHPC. The small area estimation method is used to combine a household survey and a census which should be conducted in the same year or at least close years.

## 2.2. Descriptive analysis

Electricity consumption has significantly increased in Vietnam during the 1990s and the 2000s. The percentage of households who were electrified increased from 79% in 1998 to 97% in 2010. Most households are connected to the electricity grid, and only a small number of households use other sources such as battery or pico-hydro sets (around 1%). Fig 1 indicates that the monthly per capita electricity consumption increased from 17.1 kWh in 2002 to 35.7 kWh in 2010. In line with changing lifestyles and rising ownership of durables, both the poor and non-poor households are consuming more electricity. At the macro policy level, Vietnam has implemented an incremental block structure of electricity tariff, and people have to pay higher tariffs if they consume more electricity. In spite of this, the growth rate of electricity consumption of the poor was lower than that of the non-poor. As clearly shown in Fig 1, the gaps in electricity consumption between the poor and non-poor groups have been widened over time.

Although the 2009 VHPC does not contain on electricity consumption, it has data on whether households had accessed to electricity. We can use this information to investigate the spatial distribution of the percentage of households with access to electricity. Fig 2 shows the lowest coverage of electricity in the mountainous Northern areas, followed Central Highlands and Mekong River Delta. Disaggregating down to the district level reveals a greater degree of heterogeneity in access to electricity within a province. This suggests a high inequality in electricity consumption.

## 3. Estimation method

### 3.1. Small area estimation

The main objective of this study is to examine the spatial differences in electric power consumption (kWh per capita), and then subsequently to investigate the relation between the

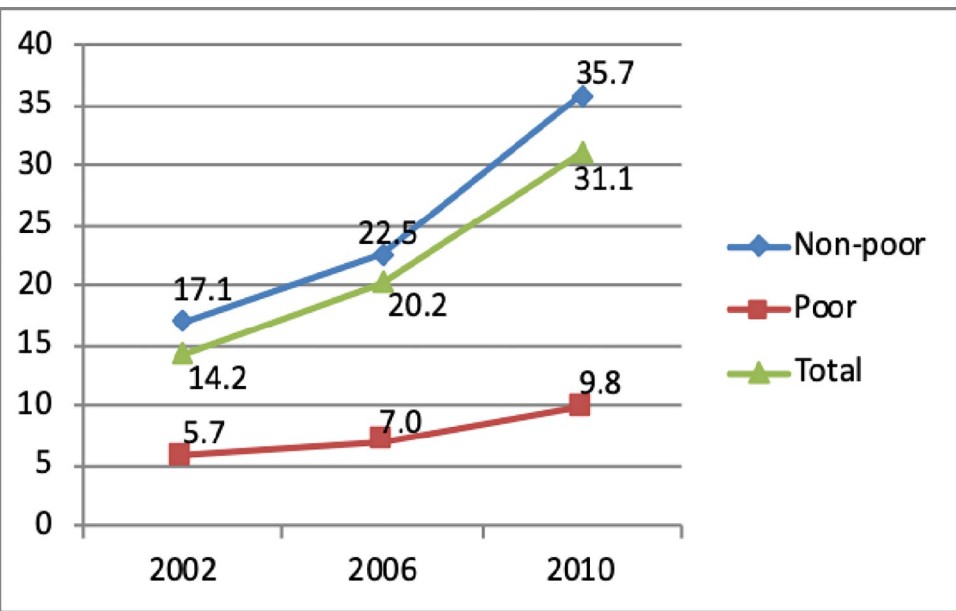

**Fig 1. Per capita electricity consumption during 2002–2010.** Source: Authors' estimation from the VHLSSs 2002, 2006 and 2010.

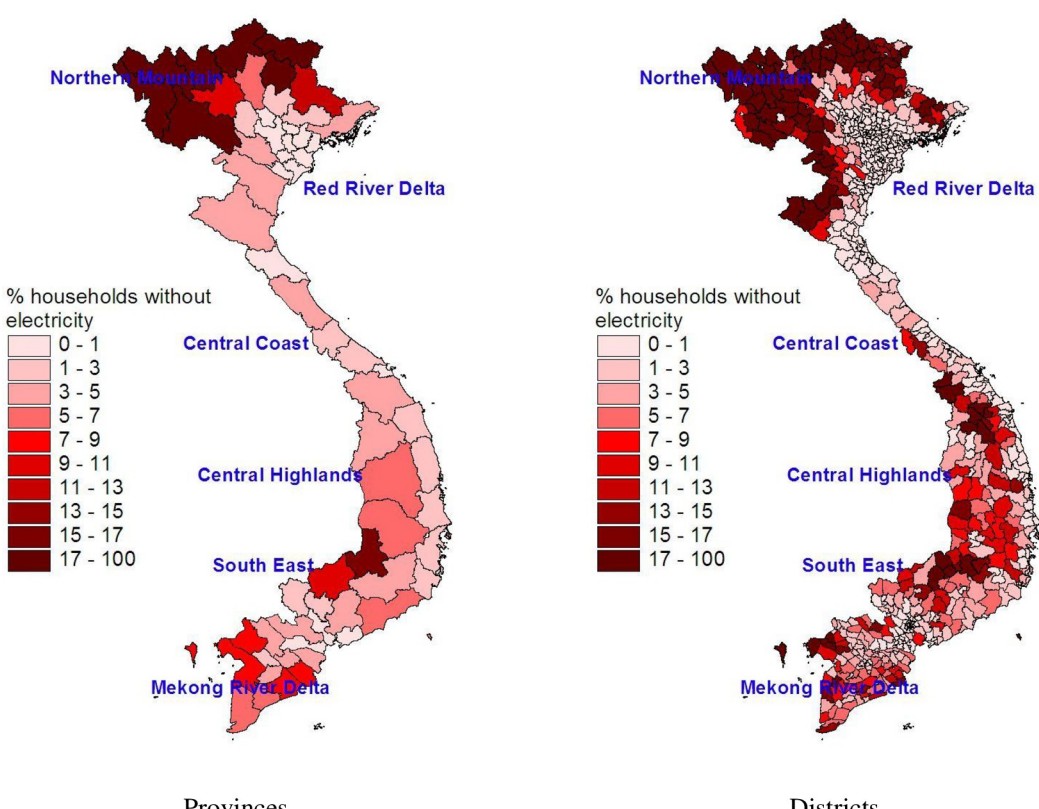

Provinces Districts

**Fig 2. Percentage of people living in a house without electricity of mainland provinces and districts.** Source: The map is drawn by the authors using data from the 2009 VPHC.

electric power consumption and economic level in Vietnam. As mentioned in the Section 2, although the 2010 VHLSS contains data on household electricity use, it is not representative at the district level due to the small number of observations. Therefore, we decided to use the small area estimation method developed by Elbers et alin order to predict the average and inequality of electric power consumption at the district level [22, 23].

Small area estimation has gained a lot of attention in recent years due to the growing need for accurate small area estimates from both public and business sectors [24–30]. Using the small area estimation method, Perseh et alestimated inequality indicators in diabetes care and health outcomes relevant to diabetes management in almost all districts of Iran [26]. Reames predicted the mean census block group home heating energy usage intensity (EUI), an energy efficiency proxy, in Kansas City, Missouri, using small area estimates approaches [31]. Using this method, numerous studies have predicted poverty estimates at the different disaggregation levels in Vietnam [27, 32] in Kenya [33], in Spain [34] and in India [35]. e Elbers et al method can be described in three steps as follows [22, 23]. In the first step, we select common variables of the 2010 VHLSS and the 2009 VPHC. The common variables include household and individual characteristics, and area mean variables computed from the census. Area mean variables are the average of household characteristics by small areas such as a commune. For example, from the census we can calculate the average age of household heads at the commune level. In the second step, we regress electricity consumption on selected common variables using data from the 2010 VHLSS:

$$\ln(kWh_{i,c,d}) = \alpha + X_{i,c,d}\beta + C_{c,d}\theta + \eta_d + \varepsilon_{i,c,d}, \tag{1}$$

where $\ln(kWh_{i,c,dd})$ is the natural logarithm of per capita monthly electricity use (measured by the number of kWh) of household $i$ in commune $c$ in a specific district. The per capita consumption is equal to the total consumption of a household divided by the number of household members. $X_{i,c,d}$ is the vector of household-level explanatory variables, and $C_{c,d}$ is the vector of the commune-level explanatory variables. The unobserved variables are decomposed into the cluster-specific random effect, denoted by $\eta_d$, and the individual-specific random effect, denoted by $\varepsilon_{i,c,d}$. This decomposition allows the correlation of errors within a cluster, i.e., district. In this study, we define districts instead of communes as clusters, since the number of households within communes is only 3, which is not efficient to estimate the within-cluster correlation.

In the third step, we apply the estimated model from the second step to the 2009 Population and Housing Census to predict the electricity consumption of households in the census:

$$\hat{\ln}(kWh_{i,c,d}) = \hat{\alpha} + X_{i,c,d}^{census}\hat{\beta} + C_{c,d}^{census}\hat{\theta} + \hat{\eta}_d + \hat{\varepsilon}_{i,c,d}, \tag{2}$$

where $\hat{\alpha}, \hat{\beta}, \hat{\theta}, \hat{\eta}_c$, and $\hat{\varepsilon}_{ic}$ denote the estimates for the corresponding parameters and error terms. These parameters and their distributions are estimated from the second step. Monte-Carlo simulations are used to estimate the point estimates as well as the standard errors of the electric power consumption of households. In each simulation, we draw a set of values of these parameters from their estimated distributions to obtain $kWh_{i,c,d}$ for all the households in the census, and then use these values to estimate the average and inequality indexes of all the districts. After $k$ simulations, we will obtain the sampling distribution of the estimates and use this to estimate the average and standard deviation of the estimates (i.e., the average and the Gini of electricity consumption at the district level).

To measure inequality, we use three common measures of inequality: the Gini coefficient, Theil's L index of inequality, and Theil's T index of inequality. These indexes have been widely used to measure inequality of income and consumption distribution. In this study, the Gini index can be calculated from the individual electricity consumption in the population [36]:

$$G = \frac{n+1}{n-1} - \frac{2}{n \cdot (n-1) \cdot k\bar{W}k_j} \sum_{i=1}^{n} \rho_j \cdot k\hat{W}h_j \tag{3}$$

where $k\hat{W}h_j$ refers to the per capita electricity consumption of individual $j$ in the 2009 Population and Housing Census that is obtained from the small area estimation. $\bar{W}$ is the average per capita electricity consumption. $\rho_j$ is the rank of person $j$ in the $Y$-distribution, counting from the person with the lowest electricity consumption to the person with the highest electricity consumption. The value of the Gini coefficient varies from zero when everyone has the same level of electricity consumption to one when one person consumes all the electric power. The closer a Gini coefficient is to one, the more unequal is the distribution of electricity consumption.

The analysis result of inequality using Theil's indexes is very similar to the result using Gini index. Since the Gini index is more widely used in measuring inequality, we use the results from the Gini index for interpretation in this paper.

## 3.2. Spatial regressions of inequality in electric power consumption

In the second step, we examine the association between the inequality in electric power consumption and economic development level of districts. Inequality is measured by the estimated Gini coefficient as shown in the subsection 3.1, while the level of economic development is measured by per capita expenditure of districts. Since the observations are

adjusted to the district level from the observations at the commune level, there can be spatial correlation between dependent variables and error terms. This possible statistical property leads us to employ a spatial model as follows:

$$Gini_d = \beta_0 + \lambda WGini_d + \beta_1 Ln(Expenditure)_d + \beta_2 Ln^2(Expenditure)_d + u_d \quad (4)$$

$$u_d = \rho M u_d + \varepsilon_d \quad (5)$$

where $Gini_d$ is the Gini coefficient of electricity consumption in district $d$. $Ln(Expenditure)$ refers to the per capita consumption expenditure in logarithmic scale. $Ln^2(Expenditure)$ is the squared log of per capita consumption expenditure, which is used to capture the potential of nonlinear relationship between power consumption inequality and economic development. $W$ and $M$ are spatial-weighting matrices (with zero diagonal elements).

In Eq (4), we regress the Gini coefficient of electricity consumption on per capita aggregate expenditure. We also regress the Gini coefficient of electricity consumption on per capita expenditure of electricity consumption. The mean and Gini coefficient of electricity consumption are estimated using the small area estimation as described above.

Since variables of districts which are closer can be spatially correlated, the above model allows for correcting the correlation of the dependent variable and error term between districts. In the presence of the model is a type of spatial econometric model with first-order spatial-autoregressive and first-order spatial-autoregressive disturbances [37–39]. The spatial-weighting matrices, $W$ and $M$, are set equal to each other and equal to the inverse-distance between centroids of districts. This matrix weighting scheme allows for taking into account the high correlation between nearby districts and for the low correlation between far districts.

## 4. Empirical results

### 4.1. Spatial distribution of the average and inequality of electricity consumption

After defining and constructing the common variables in the census and household survey, we use the 2010 VHLSS to estimate the model in Eq (1) where the log of per capita electricity consumption is related to these common variables. To the extent that there are six geographical regions in Vietnam, we estimated this model for each region. To allow for the difference in coefficients between urban and rural areas, we interact the urban variable with other explanatory variables. Forward stepwise technique is used so that only variables which are significant at least at the 5% level are kept. Only robust variables are kept in the final models. The regression results are presented in S1 to S6 Tables. We conducted two sensitive tests. Firstly, we randomly split the 2010 VHLSS into two subsamples. We estimated the electricity consumption models in the first subsample, then using these estimated models to predict electricity consumption in the second subsample and compared these results to the actual per capita expenditure. The average and inequality of electricity consumptions estimated from the small area estimation are very similar to those estimated from the actual data. Secondly, we examined the sensitivity of estimates to different specification of electricity consumption models. More specially, we estimated two models: one with a large number of explanatory variables and another with a smaller number of explanatory variables. Both models give very similar estimates of the mean and inequality of electricity consumption at the district and province levels. For interpretation in this paper, we use the estimates from the large model, since standard errors of the estimates from the large model are smaller than those from the small models.

S7 Table report the predicted per capita electricity consumption (kWh/month) and Gini coefficient of electricity consumption by provinces. Lai Chau, Ha Giang and Dien Bien are

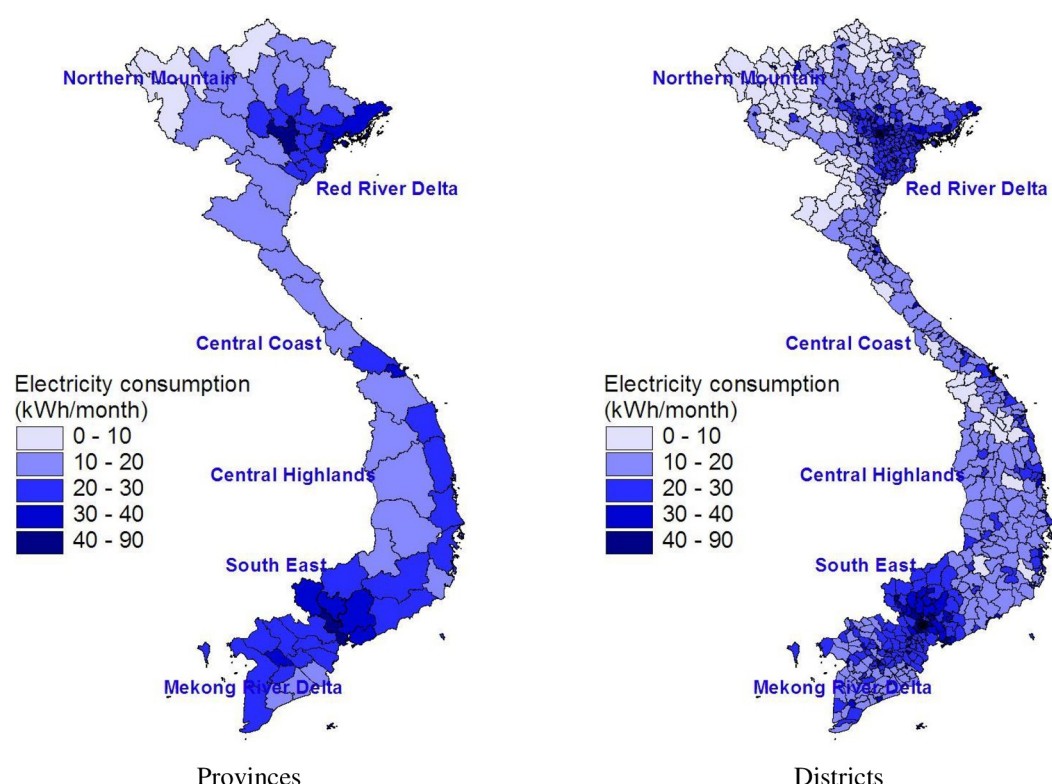

**Fig 3. Monthly kWh consumption of mainland provinces and districts.** Source: The map is drawn by the authors using data from the 2009 VPHC and the 2010 VHLSS.

three poorest provinces with the lowest consumption of electricity, which also reflects their low level of socio-economic activities. Hanoi capital and Ho Chi Minh city, which are the largest and richest cities, had the highest consumption of electricity.

The spatial consumption of electricity by provinces (left graph) and districts (right graph) is showed in Fig 3, with light blue indicating the lowest consumption level and dark blue the highest consumption level. We further note a high degree of disparity in electricity consumption within a province.

Fig 4 shows the spatial distribution of inequality in electricity consumption across provinces (left graph) and districts (right graph). The inequality seems to increase when we move from the rich provinces to the poor ones. Similarly, the regions with a low level of electricity consumption such as Northern Mountains have higher consumption inequality than other richer regions.

Table 1 presents the regional estimates of the mean and inequality of per capita electricity consumption which are computed directly from the 2010 VHLSS and those estimated from the small area estimation when using the Eqs (2) and (3). Since the 2010 VHLSS is representative at the regional level, the regional estimates based on the 2010 VHLSS can be thus regarded as the benchmark against which the estimates from the small area estimation method are compared. There is, as shown in Table 1, a similarity in the estimates between the two approaches. Regardless of the methods used, the richest region, Southeast, has the highest per capita electricity consumption, while the poorest region, Northern Mountains, has the lowest per capita electricity consumption. However, Northern Mountains has the highest inequality in electricity consumption, followed by the Mekong River Delta region.

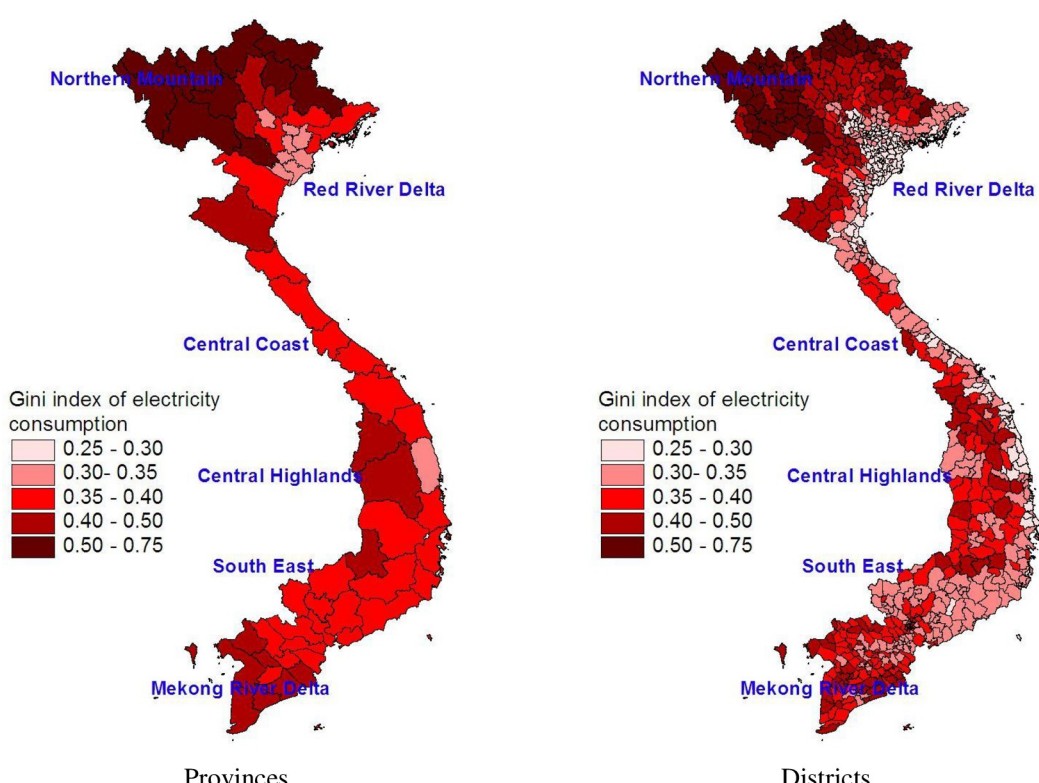

Fig 4. Gini index in monthly kWh consumption of mainland provinces and districts. Source: The map is drawn by the authors using data from the 2009 VPHC and the 2010 VHLSS.

Table 1. Regional estimates of electricity consumption.

| Regions | VHLSS 2010 | | Small area estimation | |
|---|---|---|---|---|
| | Per capita electricity consumption (kWh/month) | Gini of per capita electricity consumption | Per capita electricity consumption (kWh/month) | Gini of per capita electricity consumption |
| Northern Mountains | 16.54 | 0.481 | 16.35 | 0.527 |
| | (0.58) | (0.011) | (1.05) | (0.010) |
| Red River Delta | 36.26 | 0.425 | 34.98 | 0.391 |
| | (0.92) | (0.010) | (0.90) | (0.009) |
| Central Coast | 20.78 | 0.419 | 20.91 | 0.390 |
| | (0.47) | (0.009) | (0.70) | (0.008) |
| Central Highlands | 19.19 | 0.402 | 18.89 | 0.407 |
| | (0.77) | (0.014) | (1.75) | (0.017) |
| South East | 45.53 | 0.414 | 45.40 | 0.417 |
| | (1.56) | (0.012) | (2.44) | (0.013) |
| Mekong River Delta | 24.86 | 0.449 | 23.12 | 0.420 |
| | (0.72) | (0.010) | (0.89) | (0.008) |

Notes: Standard errors are in parentheses. The standard errors are corrected for sampling weights and cluster correlation. The estimation is based on the 2009 VPHC and the 2010 VHLSS data.

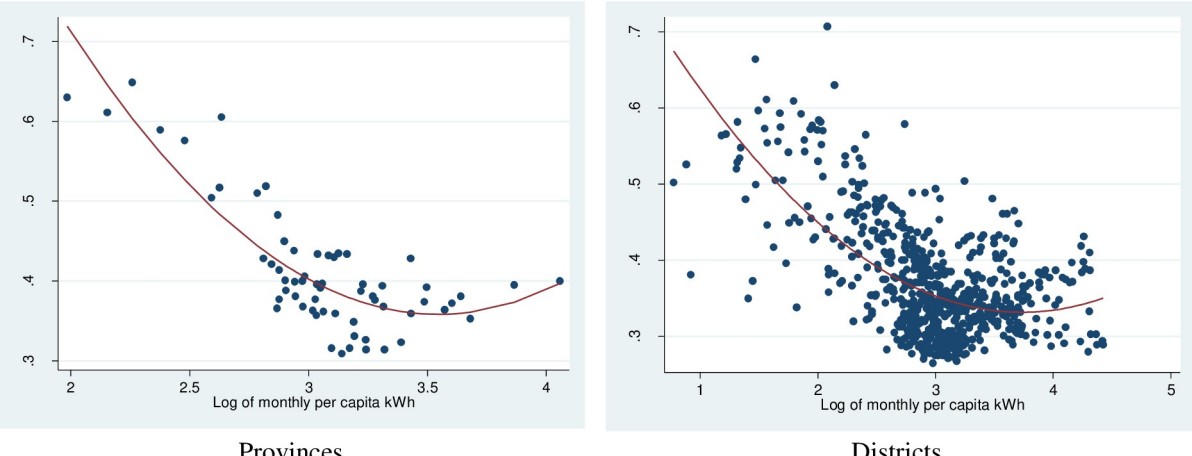

Provinces                                    Districts

**Fig 5. Gini index of monthly kWh and the average monthly kWh.** Source: Authors' estimation from the 2009 VPHC and the 2010 VHLSS.

## 4.2. Electricity consumption inequality and economic development level

Fig 5 plots together the inequality in electricity consumption and per capita electricity consumption at both provincial and district levels, where we clearly see a U-shape relationship between them. This form of relationship implies that very poor areas have very high inequality in electricity consumption. A more equal distribution is associated with a higher level of electricity consumption. However, as per capita electricity consumption continues to increase, the inequality slightly rises.

Fig 6 shows a U-shaped relationship between inequality in electricity consumption and per capita consumption expenditure (per capita total consumption expenditure of household) at the provincial and district levels. We obtained estimates of per capita aggregate consumption expenditure at the provincial and district levels from Lanjouw et al. (2016). Accordingly, the inequality in electricity consumption is reduced as the per capita consumption expenditure increases but tends to rise when the latter exceeds a certain threshold. It is also worth noting

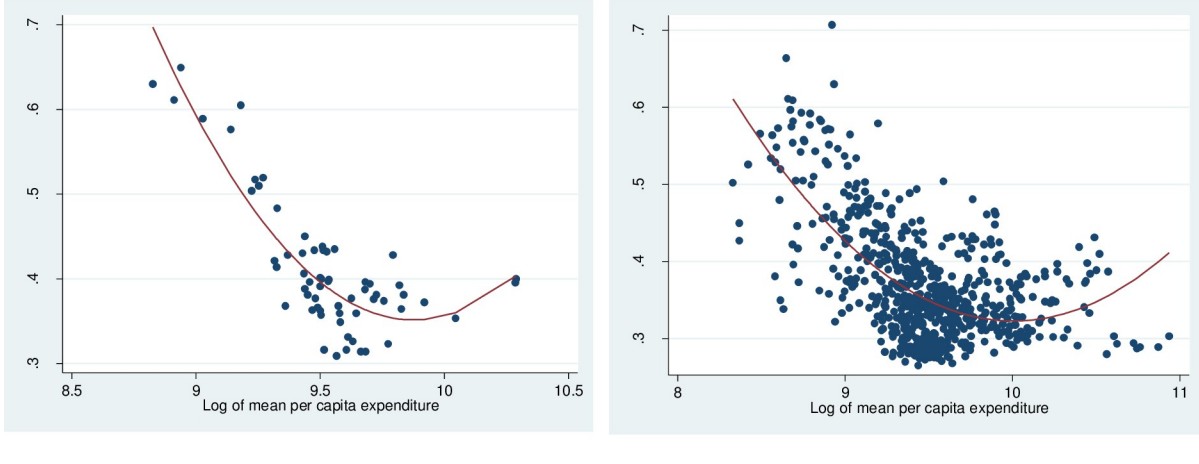

Provinces                                    Districts

**Fig 6. Gini index of monthly kWh and log of per capita expenditure.** Source: Authors' estimation from the 2009 VPHC and the 2010 VHLSS.

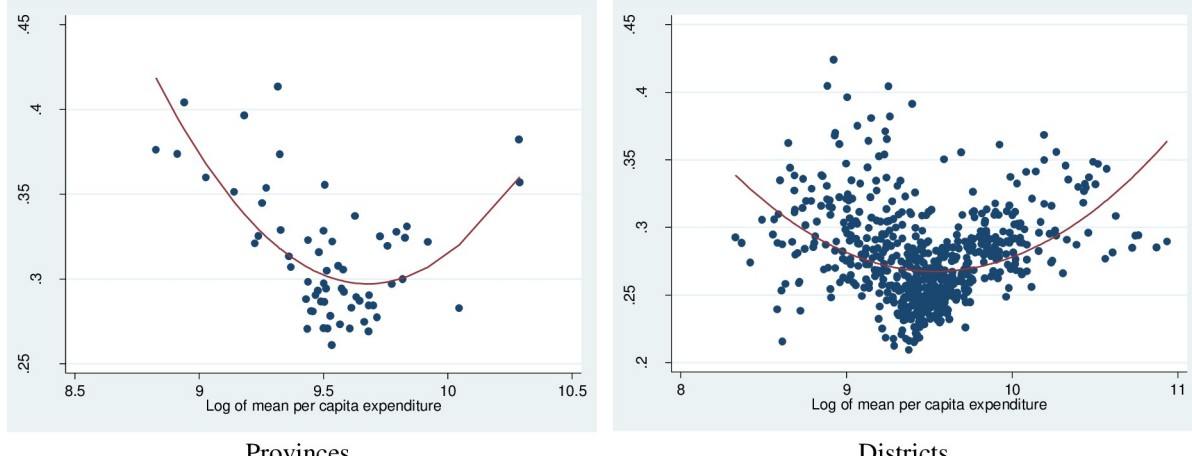

**Fig 7. Gini index of per capita expenditure and log of per capita expenditure.** Source: Authors' estimation from the 2009 VPHC and the 2010 VHLSS.

that electricity consumption inequality is strongly correlated with aggregate consumption inequality (S1 Fig), with however the latter being higher than the former. Indeed, the Gini index of aggregate consumption is 0.39, while the Gini index of electricity consumption is 0.53.

Fig 7 reports a similar relationship between inequality of per capita expenditure and per capita expenditure. This finding emphasizes that while there are certainly poor localities where everyone is similarly poor, the evidence shows that there should certainly be no presumption that inequality will be lower in poorer communities.

Table 2 reports the regressions of Gini index of electricity consumption at the district level on the level of development (measured by per capita electricity consumption and per capita household expenditure). Instead of Gini index, we also use the log of Gini index as the independent variable. The results are presented in S8 Table. The sign and significance level in the regressions of log of Gini indexes are very similar to those in the regression of Gini indexes. We conduct several robustness checks. Firstly, we try both OLS and spatial regressions, which give quite similar results. Secondly, we examine whether the estimates are sensitive to the inclusion of additional control variables. In S8 Table, we include additional control variables including regional dummies, the log of population density and the share of the urban population of districts. Overall, the model with control variables gives similar results as the model without control variables in Table 2. In this study, we aim to examine the change in electricity inequality across the economic development level instead of estimating the causal effect of economic development. Thus, we will interpret the result from regression without control variables. Thirdly, in addition to the Gini index, we use the log of the Gini index as the dependent variable. The results, reported in S9 and S10 Tables, show similar findings as those in Table 2.

We use the result from spatial regression in Table 2 for interpretation. More concretely, we find, within the spatial regressions, that the consumption inequality is significantly and negatively affected by the per capita electricity expenditure, but significantly and positively affected by the squared per capita electricity expenditure. The results thus suggest that the inequality first decreases with the per capita expenditure, but then increases when the per capital becomes higher. They also confirm the U-shape pattern that we observe in Figs 5 and 6, indicating that the power consumption inequality is highest in poor districts and lower in the middle district and then slightly increased in rich districts. Furthermore, the coefficient of weighted

**Table 2. Regression of Gini index.**

| Explanatory variables | Dependent variable is Gini index of per capita kWh consumption | | | | Dependent variable is Gini index of per capita expenditure | |
|---|---|---|---|---|---|---|
| | OLS | Spatial regression | OLS | Spatial regression | OLS | Spatial regression |
| Log of per capita expenditure | -2.0314*** | -2.3461*** | | | -0.9356*** | -0.8754*** |
| | (0.2304) | (0.1884) | | | (0.1158) | (0.0040) |
| Squared log of per capita expenditure | 0.1015*** | 0.1218*** | | | 0.0491*** | 0.0467*** |
| | (0.0120) | (0.0101) | | | (0.0061) | (0.0004) |
| Log of monthly per capita kWh | | | -0.2886*** | -0.2948*** | | |
| | | | (0.0391) | (0.0207) | | |
| Squared log of monthly per capita kWh | | | 0.0385*** | 0.0513*** | | |
| | | | (0.0063) | (0.0036) | | |
| Lambda | | -0.0053*** | | -0.0060*** | | -0.0028*** |
| | | (0.0011) | | (0.0008) | | (0.0008) |
| Rho | | 0.0704*** | | 0.0967*** | | 0.0442*** |
| | | (0.0036) | | (0.0022) | | (0.0032) |
| Constant | 10.4894*** | 11.7004*** | 0.8726*** | 0.8331*** | 4.7278*** | 4.3928*** |
| | (1.1066) | (0.8809) | (0.0596) | (0.0331) | (0.5510) | (0.0000) |
| Observations | 675 | 675 | 675 | 675 | 675 | 675 |
| R-squared | 0.4107 | | 0.4026 | | 0.1567 | |

Notes: Robust standard errors in parentheses.

\* significant at 10%

** significant at 5%

*** significant at 1%

dependent variables (Lambda) means there is a spatial negative correlation between inequality of districts. Similar findings were reported by Hasan and Mozumder who found evidence of a U-shaped relationship between income and electricity consumption in Bangladesh [40]. However, our result is only partly inconsistent with some studies in other developing countries. For instance, Ma et alfind that energy inequality in rural China decreases with higher levels of energy consumption [12], while the distribution of electricity consumption was quite equal among income group [41]. Also, electricity consumption inequality was found to increase with income levels in Indonesia [42].

## 5. Conclusion and policy implications

Vietnam has achieved high economic growth with annual GDP growth rates of around 6% in recent decades [43]. At the same time, energy consumption has also increased remarkably. According to VHLSSs, residential electricity consumption increased by 7.9% annually from 2002–2012 [20]. Although energy consumption has increased, not all households can benefit from this increase. There is a high inequality in energy consumption. In 2010, the Gini index of per capita expenditure was 0.39, while that of per capita electricity consumption was 0.52. In this study, we examine the relation between inequality in energy consumption and economic development in Vietnam.

We use a small area estimation method to estimate the average and inequality of per capita kWh consumption in Vietnam. We find that electric power consumption differs significantly across districts and provinces. Households in mountains and highlands consumed remarkable lower electricity than those living in delta and coastal areas. We find a U-shaped relation

between the inequality of electricity consumption and per capita household expenditure across districts and provinces in Vietnam. In poor districts and provinces, there is very high inequality in electricity consumption. Inequality is lower for middle income districts and provinces. The main reason for the high inequality in the poor districts is a large proportion of households without access to electricity.

Findings from our study suggest the provision of an electricity subsidy for low-income households, especially those in poor areas. The Vietnamese government has invested in the power sector for rural and mountainous areas since 2010; this has significantly improved the access to the electricity grid for a huge number of households [44]. Since 2011, the government of Vietnam has been providing cash subsidies for electricity consumption to poor households [45]. The issue of energy poverty has been significantly reduced, and inequalities in electricity consumption also declined during the 2008–2018 period [44]. However, the energy cost has recently increased, becoming a growing burden for low-income households [46]. Poor households face relatively high costs of energy, which can limit their affordability of other goods and consequently reduce their well-being [46]. Thus, electricity-subsidy policies for poor households should be further strengthened and implemented.

## Supporting information

**S1 Data. Data and do-files for journal submission.**
(RAR)

**S1 Table. GLS regressions of log of monthly per capita kWh: Northern mountains.**
(DOCX)

**S2 Table. GLS regressions of log of monthly per capita kWh: Red river delta.**
(DOCX)

**S3 Table. GLS regressions of log of monthly per capita kWh: Central coast.**
(DOCX)

**S4 Table. GLS regressions of log of monthly per capita kWh: Central highlands.**
(DOCX)

**S5 Table. GLS regressions of log of monthly per capita kWh: South East.**
(DOCX)

**S6 Table. GLS regressions of log of monthly per capita kWh: Mekong river delta.**
(DOCX)

**S7 Table. Provincial estimates of electricity consumption.**
(DOCX)

**S8 Table. Regression of Gini index with control variables.**
(DOCX)

**S9 Table. Regression of log of Gini index.**
(DOCX)

**S10 Table. Regression of log of Gini index with control variables.**
(DOCX)

**S1 Fig. Gini index of monthly kWh and Gini index of per capita expenditure.**
(DOCX)

## Acknowledgments

We would like to thank editor Carlos Alberto Zúniga-González and two anonymous reviewers from PLOS ONE for their useful comments on this study.

## Author Contributions

**Conceptualization:** Cuong Viet Nguyen, Khuong Duc Nguyen.

**Data curation:** Khuong Duc Nguyen.

**Formal analysis:** Cuong Viet Nguyen, Tuyen Quang Tran.

**Methodology:** Khuong Duc Nguyen, Tuyen Quang Tran.

**Supervision:** Cuong Viet Nguyen.

**Validation:** Cuong Viet Nguyen.

**Visualization:** Cuong Viet Nguyen.

**Writing – original draft:** Cuong Viet Nguyen.

**Writing – review & editing:** Tuyen Quang Tran.

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
