## [Decision Letter · Decision Letter 0]

13 Jan 2023

PONE-D-22-30320Inequality in Electricity Consumption and Economic Growth: Evidence from a Small Area Estimation StudyPLOS ONE

Dear Dr. tran,

Thank you for submitting your manuscript to PLOS ONE. After careful consideration, we feel that it has merit but does not fully meet PLOS ONE’s publication criteria as it currently stands. Therefore, we invite you to submit a revised version of the manuscript that addresses the points raised during the review process.

We look forward to receiving your revised manuscript.

Kind regards,

Carlos Alberto Zúniga-González, Ph.D

Academic Editor

PLOS ONE

Journal Requirements:

“The funders had no role in study design, data collection and analysis, decision to publish, or preparation of the manuscript”

5. We note that Figures 2, 3 and 4 in your submission contain map images which may be copyrighted. All PLOS content is published under the Creative Commons Attribution License (CC BY 4.0), which means that the manuscript, images, and Supporting Information files will be freely available online, and any third party is permitted to access, download, copy, distribute, and use these materials in any way, even commercially, with proper attribution. For these reasons, we cannot publish previously copyrighted maps or satellite images created using proprietary data, such as Google software (Google Maps, Street View, and Earth). For more information, see our copyright guidelines: http://journals.plos.org/plosone/s/licenses-and-copyright.

 a. You may seek permission from the original copyright holder of Figures 2, 3 and 4 to publish the content specifically under the CC BY 4.0 license. 

Additional Editor Comments:

Dear author tuyen tran, I have read your manuscript and it is interesting, but I suggest you must add a discussion section (regarding Gini and model used), where you compare your results with other authors, y section 3 or before a add a bit more literature review where you say the author that use the method of analysis. also, I would like to suggest adding more reference may be 40.

Reviewers' comments:

Reviewer's Responses to Questions

**Comments to the Author**

1. Is the manuscript technically sound, and do the data support the conclusions?

Reviewer #1: Yes

Reviewer #2: Yes

2. Has the statistical analysis been performed appropriately and rigorously? 

Reviewer #1: Yes

Reviewer #2: Yes

3. Have the authors made all data underlying the findings in their manuscript fully available?

Reviewer #1: Yes

Reviewer #2: Yes

4. Is the manuscript presented in an intelligible fashion and written in standard English?

Reviewer #1: Yes

Reviewer #2: Yes

5. Review Comments to the Author

Reviewer #1: The article adequately presents the methodology used to process the data necessary to examine the relationship between social inequity and electricity consumption. In my opinion, methodological support has been presented for the results shown; especially in those related to U-shaped relation between the inequality of electricity consumption and per capita household expenditure in Vietnamese communities

Reviewer #2: The research topic is interesting. This research is very well designed and developed.

However, I have the following comments about this study:

1. The description and use of the estimation method is not clear. The 2010 VHLSS and the 2009 VPHC are two different data sets. How to merger these data to estimate?...

2. In table 2, are the estimated results robust or not?

6. PLOS authors have the option to publish the peer review history of their article (what does this mean?). If published, this will include your full peer review and any attached files.

Reviewer #1: **Yes: **Napoleon Vicente Blanco Orozco

Reviewer #2: No

---

## [Author Response · Author response to Decision Letter 0]

2 Mar 2023

Comments from editors

Dear author Tuyen Tran, I have read your manuscript and it is interesting, but I suggest you must add a discussion section (regarding Gini and model used), where you compare your results with other authors, y section 3 or before a add a bit more literature review where you say the author that use the method of analysis. also, I would like to suggest adding more reference may be 40.

Authors: Thank you very much for this useful comment. Following you suggestion, we added more discussion comparing our findings with other studies. We also increased the number of references to 40. We added the following paragraphs. 

“Similar findings were observed by Hasan and Mozumder (2017) who found that a U-shaped relationship exists between income and electricity consumption in Bangladesh. However, our result is partly inconsistent with some studies in other developing countries. For instance, Ma et al. (2021), find in rural China that energy inequality decreases with higher levels of energy consumption. while the distribution of electricity consumption was quite equal among income group in rural China (Wu, Zheng, & Wei, 2017). Also, electricity consumption inequality was found to increase with income levels in Indonesia (Dwi Cahyani, Nachrowi, Hartono, & Widyawati, 2020).”

Following your recommendation, we also expanded the literature review and included some new references to the methods used in Section 3.1

Small-area estimation has gained a lot of attention in recent years due to the growing need for accurate small-area estimates from both the public and business sectors (Corral, Molina, Cojocaru, & Segovia, 2022; Perseh et al., 2023; Rahman, 2008). Using the small area estimation method, Perseh et al. (2023) estimated indquality indicators in diabetes care and health outcomes relevant to diabetes management in almost all districts of Iran. Reames (2016) predicted the mean census block group home heating energy usage intensity (EUI), an energy efficiency proxy, in Kansas City, Missouri, using small-area estimate approaches. Using this method, numerous studies have predicted poverty estimates at the different disaggregation levels in Vietnam ( Nguyen, 2011), Kenya (Christiaensen, Lanjouw, Luoto, & Stifel, 2012), Spain (Esteban, Morales, Pérez, & Santamaría, 2012) and India (Chandra, Aditya, & Sud, 2018).”

Reviewer 1

The article adequately presents the methodology used to process the data necessary to examine the relationship between social inequity and electricity consumption. In my opinion, methodological support has been presented for the results shown; especially in those related to U-shaped relation between the inequality of electricity consumption and per capita household expenditure in Vietnamese communities.

 Authors: Thank you very much for your comments.

Reviewer 2

The research topic is interesting. This research is very well designed and developed.

However, I have the following comments about this study:

1. The description and use of the estimation method is not clear. The 2010 VHLSS and the 2009 VPHC are two different data sets. How to merger these data to estimate?...

Authors: Thank you very much. Actually, we do not really merge the two data sets. Our main objective is to estimate the relation between inequality and mean electricity consumption at the district and province levels. The 2010 VHLSS contains data about not only expenditure on electricity but also the number of kilowatt hours (kWh) that households consumed in the last month. However, the 2010 VHLSS covered 9,399 households. It is just a sampled survey, which is not representative for provinces and districts. On the other hand, the 2009 VPHC has a large coverage of households, and it is representative at the provincial and district levels, but it does not contain data on electricity consumption of households. We first use the 2010 VHLSS to construct a model of electricity consumption as a function of household and community variables that are are available in both the 2010 VHLSS and the 2009 VHPC. Then, the parameter estimates from this model are applied to the 2009 VHPC to predict electricity consumption of all households in the population. These household-level data allow use to estimate the mean and inequality of electricity consumption for provincial and districts.

In the revised manuscript, we added a paragraph in the Introduction section to note the data combination. 

2. In table 2, are the estimated results robust or not?

Authors: Thank you very much for this useful comment. Following you suggestion, we conducted additional robustness analyses. Firstly, we try both OLS and spatial regressions, which give quite similar results. Secondly, we examine whether the estimates are sensitive to inclusion of additional control variables. In Table A.8 in the Appendix, we include additional control variables including regional dummies, log of population density and the share of urban population of districts. Overall, the model with control variables gives similar results as the model without control variables in Table 2. In this study, we aim to examine the change in electricity inequality across the economic development level instead of estimating the causal effect of the economic development. Thus, we will use the result from regression without control variables for interpretation. Thirdly, in addition to the Gini index, we use log of the Gini index as the dependent variable. The results, which are reported in Tables A.9 and A.10 in the Appendix, show similar findings as those in Table 2.

---

## [Decision Letter · Decision Letter 1]

15 Mar 2023

PONE-D-22-30320R1Inequality in Electricity Consumption and Economic Growth: Evidence from a Small Area Estimation StudyPLOS ONE

Dear Dr. Tran,

Thank you for submitting your manuscript to PLOS ONE. After careful consideration, we feel that it has merit but does not fully meet PLOS ONE’s publication criteria as it currently stands. Therefore, we invite you to submit a revised version of the manuscript that addresses the points raised during the review process.

We look forward to receiving your revised manuscript.

Kind regards,

Carlos Alberto Zúniga-González, Ph.D

Academic Editor

PLOS ONE

Journal Requirements:

Additional Editor Comments:

Dear authors, thanks for your contribution for improvements your manuscript, so for finalize the editorial process only make the improvement indicated for reviewers 2.

Reviewers' comments:

Reviewer's Responses to Questions

**Comments to the Author**

1. If the authors have adequately addressed your comments raised in a previous round of review and you feel that this manuscript is now acceptable for publication, you may indicate that here to bypass the “Comments to the Author” section, enter your conflict of interest statement in the “Confidential to Editor” section, and submit your "Accept" recommendation.

Reviewer #1: All comments have been addressed

Reviewer #2: All comments have been addressed

2. Is the manuscript technically sound, and do the data support the conclusions?

Reviewer #1: Yes

Reviewer #2: Yes

3. Has the statistical analysis been performed appropriately and rigorously? 

Reviewer #1: Yes

Reviewer #2: Yes

4. Have the authors made all data underlying the findings in their manuscript fully available?

Reviewer #1: Yes

Reviewer #2: Yes

5. Is the manuscript presented in an intelligible fashion and written in standard English?

Reviewer #1: Yes

Reviewer #2: Yes

6. Review Comments to the Author

Reviewer #1: From my point of view, the author has incorporated all the observations made in the previous review and has done so objectively and purposefully.Therefore, I consider that this work should be published.

Reviewer #2: From 2010 to now, the Government has invested in the energy sector for mountainous areas. The problem of energy poverty in Vietnam has also been improved significantly. So, I want the authors to further clarify the policy implications associated with the Vietnamese context.

7. PLOS authors have the option to publish the peer review history of their article (what does this mean?). If published, this will include your full peer review and any attached files.

Reviewer #1: **Yes: **Napoleon Vicente Blanco Orozco

Reviewer #2: No

---

## [Author Response · Author response to Decision Letter 1]

18 Mar 2023

Reviewer #2: From 2010 to now, the Government has invested in the energy sector for mountainous areas. The problem of energy poverty in Vietnam has also been improved significantly. So, I want the authors to further clarify the policy implications associated with the Vietnamese context.

Authors’ responses:

Thank you so much for your helpful comments. Following your suggestion, we have addressed this implicatin in the revision as follows:

Findings from our study suggest the provision of an electricity subsidy for low-income households, especially those in poor areas. The Vietnamese government has invested in the power sector for rural and mountainous areas since 2010; this has significantly improved the access to the electricity grid for a huge number of households (Minh and Nguyen, 2021). Since 2011, the government of Vietnam has been providing cash subsidies for electricity consumption to poor households (Government of Vietnam, 2011). The issue of energy poverty has been significantly reduced, and inequalities in electricity consumption also declined during the 2008–2018 period (Minh et al., 2021). However, the energy cost has recently increased, becoming a growing burden for low-income households (Nguyen, Nguyen, Hoang, Wilson, & Managi, 2019). Poor households face relatively high costs of energy, which can limit their affordability of other goods and consequently reduce their well-being (Nguyen et al., 2019). Thus, electricity-subsidy policies for poor households should be further strengthened and implemented.

---

## [Editor Report · Decision Letter 2]

22 Mar 2023

Inequality in Electricity Consumption and Economic Growth: Evidence from a Small Area Estimation Study

PONE-D-22-30320R2

Dear Dr. tuyen tran,

We’re pleased to inform you that your manuscript has been judged scientifically suitable for publication and will be formally accepted for publication once it meets all outstanding technical requirements.

Kind regards,

Carlos Alberto Zúniga-González, Ph.D

Academic Editor

PLOS ONE

Additional Editor Comments (optional):

Dear author I have checked that you have been the corrections and clarified the observations of reviewer # 2. My sincere congratulations, my decision is accepted.
---

## [Editor Report · Acceptance letter]

10 Apr 2023

PONE-D-22-30320R2 

Inequality in Electricity Consumption and Economic Growth: Evidence from a Small Area Estimation Study 

Dear Dr. Tran:

I'm pleased to inform you that your manuscript has been deemed suitable for publication in PLOS ONE. Congratulations! Your manuscript is now with our production department. 

Kind regards, 

on behalf of

Dr. Prof. Carlos Alberto Zúniga-González 

Academic Editor

PLOS ONE